# Space-time optical diffraction from synthetic motion

A. C. Harwood [1,5] ✉, S. Vezzoli [1,5], T. V. Raziman [1,2], C. Hooper [3], R. Tirole [1], F. Wu[1], S. A. Maier [1,4], J. B. Pendry [1], S. A. R. Horsley [3] & R. Sapienza [1] ✉

The interaction of light with objects and media moving at relativistic and superluminal speeds enables unconventional phenomena such as Fresnel drag, Hawking radiation, and light amplification. Synthetic motion, facilitated by modulated internal degrees of freedom, enables the study of relativistic phenomena unrestricted by the speed of light. In this study, we investigate synthetically moving apertures created by high-contrast reflectivity modulations, which are generated by ultrafast laser pulses on a subwavelength thin film of indium tin oxide. The space-time diffraction of a weaker probe beam reveals a complex, non-separable spatio-temporal transformation, where changes in the frequency of the wave are correlated to changes in its momentum. By using schemes of continuous or discrete modulation we demonstrate tunable frequency-momentum diffraction patterns with gradients that depend upon the relative velocity between the modulation and the probe wave. The diffraction patterns are matched by operator-based theory and the gradients are analytically predicted using a super-relativistic Doppler model, where the modulation is described as a superluminally moving scattering particle. Our experiments open a path towards mimicking relativistic mechanics and developing complex and programmable spatio-temporal transformations of light.

Diffraction from a structured surface endows many natural materials with a striking appearance and is currently harnessed in applications from displays to spectroscopy. Taking this effect to its limit, spatial metasurfaces[1] have revolutionised the shaping of light waves with high-contrast, subwavelength nanostructures. Yet, spatial structuring of the dielectric permittivity $\varepsilon(x)$ can only ever change a wave's momentum content, leaving its energy (frequency) distribution untouched (Fig. 1a, brown line).

Conversely, time-varying metamaterials harness a permittivity that is structured on sub-period timescales, $\varepsilon(t)$, allowing them to transform the frequency content of interacting waves, while conserving momentum (Fig. 1a, red line)[2,3]. This promise of spectral control has been brought to the optical regime by the very large ultrafast

modulations of permittivity enabled by high intensity, pulsed lasers[4,5], leading to the development of time-varying metasurfaces. By combining traditional nano-structuring in space with optically fast modulations of the dielectric properties it is possible to achieve separable space-time structuring of the form $\varepsilon(x, t) = \varepsilon(x) \times \varepsilon(t)$, which can transform the frequency $\omega$ and wavevector $\mathbf{k}$ of the light[6,7].

Non-separable space-time permittivity structures, $\varepsilon(x, t) \neq \varepsilon(x) \times \varepsilon(t)$, enable in-principle arbitrarily complex space-time transformations of an incident optical field and thus the synthesis of any light spectrum $E(\mathbf{k}, \omega)$ (Fig. 1b), as well as non-reciprocal photonic transitions between different momenta and energy[8–10]. The simplest example of a non-separable modulation is of the form $\varepsilon(x, t) = \varepsilon(x - vt)$, where $v$ is the synthetic velocity of the modulation[11]. This apparent velocity

[1]Blackett Laboratory, Department of Physics, Imperial College London, London SW7 2AZ, UK. [2]Department of Mathematics, Imperial College London, London SW7 2BW, UK. [3]School of Physics and Astronomy, University of Exeter, Exeter EX4 4QL, UK. [4]School of Physics and Astronomy, Monash University, Clayton, VIC 3800, Australia. [5]These authors contributed equally: A. C. Harwood, S. Vezzoli. ✉e-mail: a.harwood22@imperial.ac.uk; r.sapienza@imperial.ac.uk

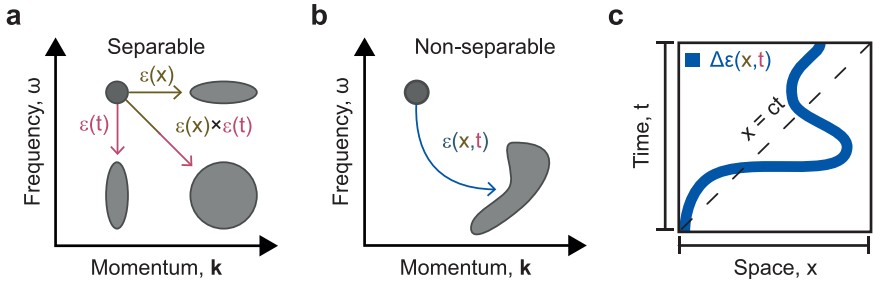

**Fig. 1 | Space-time diffraction from a synthetically moving modulation.**
**a** Separable modulations of the dielectric permittivity ε can only induce transformations of the optical spectrum parallel to either the frequency or momentum axes. **b** Meanwhile, the general non-separable modulations of ε(x, t), demonstrated here, can perform arbitrary momentum-frequency transformations that are unobtainable with static metasurfaces. **c** The synthetic motion of permittivity modulations, Δε(x, t), can describe arbitrary trajectories in space-time (solid blue line), even moving at superluminal velocities outside the light cone.

arises with zero inertia, akin to the motion of a laser spot on a distant screen, and therefore may be superluminal. Moreover, modulations that display complex synthetic motion of the form $\varepsilon(x, t) = \varepsilon(x - f(t))$ (Fig. 1c, solid blue line) can describe any conceivable trajectory.

The interactions between light and synthetically moving modulations, induced by pulses propagating in nonlinear fibres or bulk crystals, has been used for analogue simulations of black holes[12–14] and gravitational effects[15]. These systems rely upon long propagation distances to accumulate significant signal, due to the perturbative nature of the modulation, limiting the ability to control space and time independently and generate complex motions. In contrast, a space-time metasurface featuring arbitrarily complex, high-contrast modulations—optically induced by pulses independently structured in both space and time—would provide an exciting platform for mimicking relativistic mechanics[16] or generating novel entangled states[17].

In this letter we showcase two schemes for generating optically induced, non-separable space-time modulations of the simple form $\varepsilon(x - vt)$, which behave as highly efficient diffractive elements and represent the building blocks towards a general space-time metasurfaces. We first demonstrate the continuous synthetic motion generated by a broad, obliquely incident laser pulse, followed by the discrete synthetic motion induced by localised pumping at different points in space and time. By probing these synthetically moving, ultrafast modulations with a weak probe beam, we observe diffraction both in momentum and frequency. These space-time diffraction patterns are closely matched by an operator-based numerical theory and an analytical theory of super-relativistic Doppler shift. Our results promise to lead the way towards programmable optical potentials for analogue computing and simulations of relativistic phenomena.

## Results

We generate travelling space-time modulations of the permittivity $\varepsilon(x, t)$ in a deeply subwavelength film of Indium Tin Oxide (ITO), induced by intraband photocarrier excitation with high intensity, ultrafast laser pulses[18–20]. Changes in the permittivity of ITO about the medium's epsilon-near-zero (ENZ) wavelength in the near infrared are observed as a large temporal modulation of the complex reflection coefficient $r(x, t)$ of the film, localised around the position of the pump beam in space-time[5,21]. Experimentally, we use a sample comprised of a 40 nm film of ITO (ENZ frequency of 227 THz, 1320 nm) sandwiched between a glass coverslip and a 100 nm gold layer used to improve field confinement and reflectivity modulation contrast[22] (see Fig. 2a). Excitation with 225 fs, ~200 GW/cm² (~2 μJ) pump pulses (centred at 231 THz, or 1300 nm), induces a time-varying reflectivity $R = |r|^2$ from few % up to ~70%, with short rise time (down to few optical cycles, <10 fs, for high pump powers) and longer decay (~ 200 fs)[18,21–23].

These reflectivity modulations are then investigated by an angularly distinguishable, spectrally degenerate probe (green line in Fig. 2a), which has been stretched in time by using a top-hat spectral filter (~ 1 THz, see SI for more details). The probe beam has intensity low enough to not significantly alter the material's reflectivity and induces a linear polarisation wave inside the ITO. The probe light diffracts from the reflectivity modulation, which is generated using pump pulses that arrive at an angle $\theta_i$ (red lines) with a controllable time delay $\delta t$. We characterise the space-time diffraction pattern of the probe beam using hyperspectral measurements, capturing the frequency spectrum of the scattered light at a selection of angles ($\delta\theta$) around the probe beam's angle of reflection, as illustrated in Fig. 2a (see Supplementary Fig. 1 for details on the setup).

### Continuous synthetic motion

Our first study concerns space-time diffraction from modulations that travel across the surface of the sample with a continuous synthetic velocity $v_r$. Herein, we consider motion in one spatial dimension, $x$. By using a single spatially stretched pump beam that is obliquely incident at an angle $\theta_i$, we generate a modulation of the sample's reflection coefficient $r(x, t)$ that travels at velocity $v_r = c/\sin(\theta_i)$ across the sample surface (as sketched in Fig. 2d, e). The simplified shape of the space-time reflective slit is here sketched as a parallelogram for clarity (blue in the space-time diagram shown in Fig. 2b), but instead depends on the temporal response of the ITO/Au bilayer to the pump excitation (see Supplementary Fig. 4). Likewise, using an obliquely incident broad probe beam, we generate a second linear polarisation wave that moves with synthetic velocity $v_p$ (dashed green line in Fig. 2b). To probe the modulation's complete dynamic, we stretch the probe beam in space by using a physical slit in the Fourier plane of the focusing lens (see SI), to around three times the pump extension, as well as in time, to a duration of 690 fs. By consequence, the probe fully encapsulates the modulation in space-time, resulting in the green ellipse of Fig. 2b. We remark that this is the opposite configuration of a conventional pump-probe setup, where the pump is spatially very large, and the probe is much narrower. Indeed, in the y direction the pump is larger than the probe beam and no diffraction is observed.

The travelling, non-perturbative modulations of both the amplitude ($\rho$) and phase ($\phi$) of the reflection coefficient, $r = \rho\, e^{j\phi}$, diffracts light in both space and time, leading to correlated broadening and shifts in both the wavevector (momentum) and frequency spectra, representing the space-time generalisation of the law of refraction[24–26]. Maximal spatio-temporal diffraction occurs when the probe and the reflectivity modulation have temporal overlap on the surface. This happens slightly before the zero delay between the pump and probe pulses, as shown in Fig. 2c, due to the asymmetric time response of ITO. The diffraction efficiency is defined as the signal detected outside of the spectral and angular bandwidth of the incident probe (red box in central panels of Fig. 2d, e, top-hat shaped by design), normalised to the total input signal without modulation. Diffraction efficiency reaches an experimental maximum of 4.2%, however, it is important to stipulate that due to the stretching of the probe in space and time, only 19% of the probe is

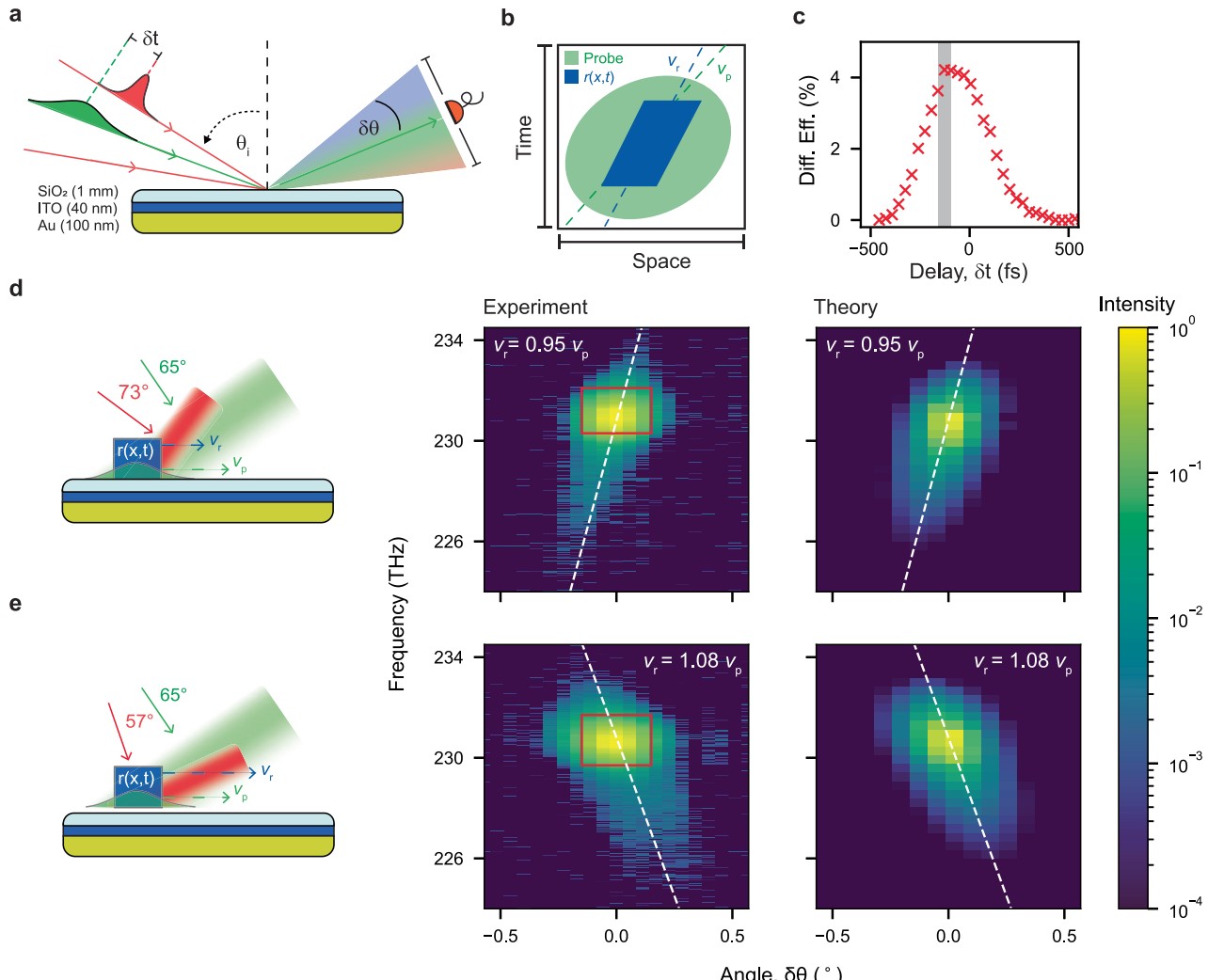

**Fig. 2 | Synthetic motion from a single, continuous reflectivity modulation.**
**a** We study space-time diffraction from synthetically moving reflectivity modulations using a pump-probe experiment, featuring one of two pump beam lines (red) and a central probe line (green) with non-degenerate angles of incidence $\theta_i$, determining synthetic velocities on the surface. The space-time diffraction of the probe beam is measured by the angular dependence ($\delta\theta$) of the spectrum diffracted about the reflected probe. **b** Sketches of the space-time profile of an idealised moving reflectivity slit (blue) and of the probe beam (green), along with their synthetic velocities (dashed lines). **c** The experimental diffraction efficiency (red crosses) of the probe from a single moving modulation as a function of pump-probe delay $\delta t$, defined as the ratio of the scattered to the initial, unmodulated probe power. **d**, **e** The frequency-momentum hyperspectral plots of the diffracted probe power density for the case of a slit that travels slower ($v_r = 0.95 \, v_p$, top row) and faster ($v_r = 1.08 \, v_p$, bottom row), than the probe, taken at the delay of maximum efficiency (shaded grey region in **c**). The sketches illustrate the elongated nature of both pump and probe beams, their relative size, and their incident angles, leading to measurable signatures of synthetic motion. The correlation between frequency and momentum depends on the relative speed of probe and modulation, in excellent agreement with analytical theory and numerical simulations (white dashed lines and colormaps on the right).

spatio-temporally overlapped with the modulation. The theoretical maximum achievable diffraction efficiency for our experimental conditions, given by space-time diffraction from a modulation with perfect contrast, enables a comparable efficiency of 10%.

Figure 2d, e shows the experimental and theoretical frequency-momentum resolved diffraction of a probe wave of constant synthetic velocity $v_p$, scattering from travelling modulations of different relative velocities, either slower ($v_r = 0.95 \, v_p$, panel d) or faster ($v_r = 1.08 \, v_p$, panel e) than the probe ($v_p = 1.10 \, c$). All synthetic velocities used in this experiment are inherently superluminal. The ultrafast temporal dynamic of the modulation of the reflectivity induces a spectral broadening and red-shift of the incident probe light (top-hat incident spectrum centred at 231 THz with a bandwidth of 1 THz, red box in Fig. 2d, e) with new frequencies extending between 225 THz and 234 THz. The spectral extension of the diffracted light exhibits a characteristic red-shift, which has been previously observed in time-only experiments[5,20,21]. This asymmetric spectral shift arises from the

ultrafast evolution of the phase of the modulated reflection coefficient, which has a significantly faster rise time ($\sim 10$ fs), inducing a large red-shift, than its decay time ($\sim 200$ fs), inducing a much smaller blueshift.

The theoretical predictions (Fig. 2d, e) are calculated from an operator theory with modulation of the plasma frequency in ITO[27], extended to include both spatial and temporal diffraction (more details in SI). The main feature in the hyperspectral data is the clear correlation between diffracted angle and frequency, which switches direction with the sign of the relative velocity between the modulation and the probe wave. The gradient that describes this correlation is captured by an analytical simplification of the operator theory model (see SI for the derivation):

$$\delta\omega \propto \left(1 - \frac{v_r}{v_p}\right)^{-1} \delta\theta. \qquad (1)$$

The linear relationship given by Eq. (1) is plotted in Fig. 2d, e as a white dotted line over both the experimental and theoretical diffraction patterns, demonstrating remarkable agreement. Additionally, we note that the dependency of the gradient upon the relative velocity of the modulation is a signature of non-reciprocity, i.e., a probe sent from the diffraction direction with inverted temporal evolution (propagating backwards in time) would not retrace the same path and would emerge with different $\mathbf{k}$ and $\omega$[28]. Non-reciprocity is inherent in all systems that break time translation symmetry, however, by utilising the ultrafast response of ITO, our system extends past research into non-reciprocal optical metasurfaces[8] to encompass a greater bandwidth. Non-reciprocal integrated devices with near optical frequency bandwidth would enable frequency and spatial multicasting and multiplexing and are coveted in the fields of LiFi, LiDAR, and space communication[29].

Furthermore, the gradient predicted by the operator theory model can be analytically calculated by describing the moving modulation as a relativistic particle with superluminal velocity that scatters the probe light according to the relativistic Doppler effect (see SI for derivation). The hyperspectral gradient and sign flip obtained using this relativistic Doppler effect model perfectly matches Eq. (1), following the paradigm of an inverse Doppler shift, in which longer wavelengths of light are observed in scatterer's direction of motion and shorter wavelengths are scattered in the opposite direction.

### Discrete synthetic motion

As shown so far, an obliquely incident beam induces a travelling modulation with constant velocity as it continuously intersects the sample surface at different positions and times, acting as a non-separable space-time potential or transformation for light. A second way to define a travelling modulation, which in principle allows for the design of arbitrary space-time trajectories, is through a set of discrete modulations concentrated at space-time points $(x_i, t_i)$. These can be defined through multiple excitations with controlled relative separations in space ($\delta x$) and delays in time ($\delta t$), as shown in Fig. 3a for the simple case of a double excitation. These define a discrete effective velocity $v_r = \delta x/\delta t$, as the slope of the line connecting the reflectivity modulations $r(x_1, t_1)$, and $r(x_2, t_2)$, illustrated by the blue patches and the connecting dashed blue line in Fig. 3b (the green line represents probe velocity $v_p$, which is the same as before).

To explore this discretised scheme of generating synthetic motion, we use tightly focused pump beams, rather than extended ones, whose individual motion over such short distance can be neglected (see SI for a comparison). Similar to the case of continuous motion, here we use a probe beam that is again stretched in space and time encompassing the complete synthetic space-time trajectory (its space-time power distribution is sketched in Fig. 3b). As the modulated area, and consequently the diffraction efficiency decreases, we acquire the frequency-momentum hyperspectra away from $\delta\theta = 0°$, thus eliminating the strong signal of the undiffracted probe. In analogy to the space-only and time-only cases ($\delta t = 0$ and $\delta x = 0$, respectively), the discreteness of the modulation leads to interference fringes in the frequency-momentum spectrum, whose period is inversely related to the distance the modulation patches in space-time. This is the spatio-temporal version of Young's double-slit experiment, which generalises both the spatial and the temporal versions[5] of this classic experiment.

The non-separable coupling of frequency and momentum is evident in the diagonal interference fringes, which are observed in the diffraction patterns (Fig. 3c), and well captured by the numerical theory (Fig. 3d). By comparison, we note that a separable spatial and temporal modulation would induce a checkerboard pattern in frequency-momentum space, and a criss-crossing set of fringes (see SI). Alike to the non-separable continuous motion of Fig. 2, the sign of the slope of the fringes depends only on the relative velocity between the probe and the effective velocity $v_r/v_p$, and not on the superluminal

or subluminal nature of the motion (top row is superluminal, bottom row subluminal). Moreover, in this scheme, the synthetic speed of the motion, as well as the relative distance of the space-time slits may be continuously tuned via their relative separation and delay, as illustrated by Fig. 3b. This gives access to a large range of synthetic velocities (see SI for more examples) and represents the first step towards the construction of arbitrary space-time trajectories.

## Discussion

The results presented here establish a framework for generating tunable momentum-frequency beam profiles through non-separable space-time diffraction from an ultrafast modulation moving with synthetic velocities, which extend beyond the speed of light. The ability to generate these novel hyperspectral diffraction patterns stems from the large and ultrafast changes in both amplitude and phase of the complex reflectivity of ITO close to the ENZ frequency, which may be controlled via the intensity of the pump pulse[21]. Detection of this diffraction is only possible using the correct angular, spatial, and temporal laser beams, as in the bespoke pump-probe setup employed here, conditions which are not usually met in conventional pump and probe experiments. The ability to access both phase and amplitude modulations, in conjunction with the demonstrated high diffraction efficiency, allows for the creation of complex and asymmetric momentum-frequency spectra, beyond Friedel's law[30,31] which dictates that the scattering amplitude is symmetric and is the Fourier transform of the permittivity modulation, $\delta\varepsilon \sim (\Delta\mathbf{k}, \Delta\omega)$. Moreover, the simple space-time modulations demonstrated here can be extended, by using complex pump beams that are structured in both space and time, to realise programmable and complex spatio-temporal transformations, surpassing what can be achieved by sequences of purely spatial or purely temporal modulation[32].

The construction of apparent motion using discrete, stationary modulations can be extended to study exotic and super-relativistic kinds of synthetic motions, as illustrated in Fig. 1c. This prospect of space-time control of light greatly improves upon conventional perturbative nonlinear optical systems, where interactions with low contrast (change in refractive index ≪ 1) modulations require long propagation distances to accrue significant signal—a process that introduces deleterious signals and restricts the control of space and time independently, e.g., as in a 1D fibre. Furthermore, a planar optical device that may be readily structured in two spatial dimensions, as well as in time, invites the expansion of our scheme to the generation of spatio-temporally structured light[33] and the study of novel rotational motion schemes[11].

Finally, our experimental results are supported by an operator-based numerical theory that demonstrates excellent agreement with the shape and gradient of diffraction patterns, demonstrating the validity of this approach in describing time-varying systems. Yet, we note that this model is unable to fully capture the spectral extent of the experimental results, indicating that the temporal dynamic of the change in reflectivity is much faster than predicted, a finding consistent with previous studies[21,22]. The relative velocity-dependent gradient of the diffraction patterns is interpreted as the super-relativistic Doppler effect, demonstrating the potential of our system for studying novel relativistic phenomena through optical simulations. Whereas past optical, analogue gravity experiments used the interplay of dispersion and modulations to create and study effective horizons[15,34], the ability to generate high-contrast effective horizons with readily tunable velocities described herein offers a potent alternative for studying analogue relativistic phenomena[35,36].

In conclusion, we have shown that a synthetically moving reflectivity modulation can be generated by either continuous or discrete illumination of a subwavelength film of ITO and how it can spatio-temporally diffract incident probe light with very high efficiency. We

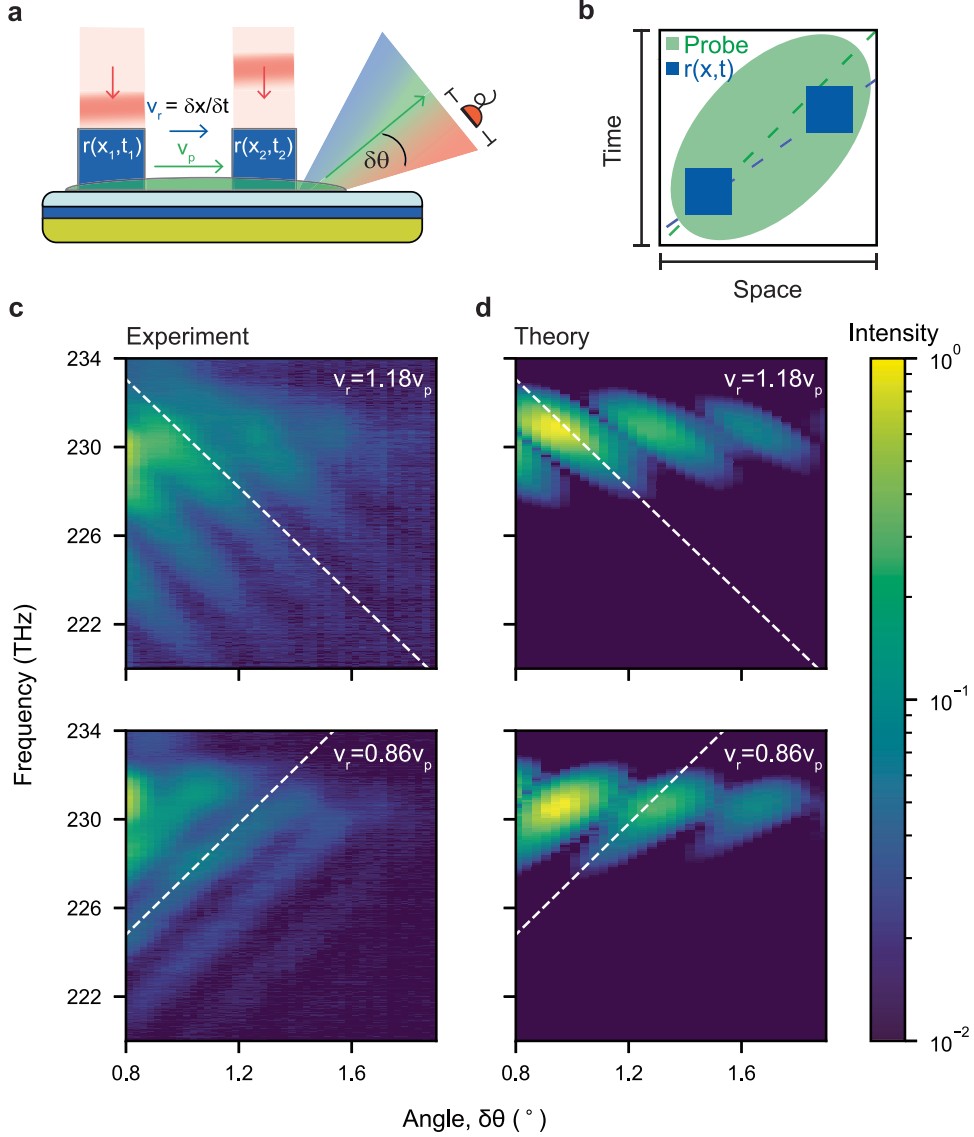

**Fig. 3 | Frequency-momentum interference from a pair of reflectivity modulations. a** Discrete synthetic velocity is generated when two localised pump pulses are incident on the surface of the ITO with a tuneable spatial separation ($\delta x$) and temporal delay ($\delta t$), inducing an effective reflectivity modulation travelling at synthetic velocity $v_r = \delta x/\delta t$. This is the generalisation of the Young's double-slit experiment to space-time. **b** Sketch of the spatio-temporal profile of the two pumps (blue patches) and of the probe (green ellipse). The dashed line represents their respective synthetic velocities. **c**, **d** Experimental and theoretical results for the case $v_r = 0.86\, v_p$, and for the case $v_r = 1.18\, v_p$. Clear, diagonal diffraction fringes are visible which extend much further than the input frequency and momentum bandwidth of the probe (230.5 to 231.5 THz and 64.8 to 65.2°, respectively). The slope of the fringes depends on the ratio $v_r/v_p$, like before, whereas their period depends on the slit separation in space-time.

observe a correlated diffraction in frequency and momentum, whose gradient depends on the relative velocity between the probe and the modulation, as simulated by an operator-based numerical model and in agreement with a super-relativistic Doppler effect. The possibility to engineer complex synthetic motions at optical frequencies, represents a powerful tool to explore fundamental physical phenomena that are inaccessible in the laboratory or impossible due to physical constraints[11]. Moreover, the development of high-contrast space-time diffractive elements is a crucial step towards the construction of a space-time metasurface, capable of harnessing non-Hermitian spatio-temporal phenomena such as photonic drag[37], Doppler cloaking[38], travelling-wave amplification[39] and instabilities[40], analogue Hawking radiation[41] and dynamic Casimir effects[16,18], among many others. Critically, such a device offers broad applications across optical devices, with impacts in sensing, communications and computing[42,43], imaging and augmented reality, beam shaping and fast switching[3,33].

## Methods

### Sample

The sample is a 40 nm film of Indium Tin Oxide (ITO) (from Präzisions Glas & Optik GmbH) with an epsilon-near-zero frequency of 227 THz (1320 nm), deposited on a substrate of glass and covered by a 100 nm layer of gold. The gold layer enhances the field confinement and amplifies the reflectivity modulation (measured about 230.2 THz/ 1300 nm) induced by changes in the complex refractive index.

### The optical setup

We utilise an angularly resolved pump-probe setup (Supplementary Fig. 1a), which manipulates ultrafast, near-infrared pulses (230.2 THz/ 1300 nm, 225 fs full-width at half-maximum) generated by a PHAROS (Light Conversion) solid-state laser coupled with an ORPHEUS (Light Conversion) optical parametric amplifier. The laser light from the optical parametric amplifier is separated into three beam paths,

corresponding to the lower intensity probe line (green) and the two higher intensity pump lines (red), via beam splitters. These three paths are incident upon the sample from three separate angles, 65° for the probe beam and 57° and 73° for the two pump beams, referred to throughout this document as 'pump 1' and 'pump 2' respectively. Due to the angularly broad Berreman resonance hosted by our sample, the two pump beams modulate the permittivity of the sample with similar efficiency. Two delay stages in the paths of pump 2 and the probe are then used to synchronise the arrival times of pulses from the three paths at the sample, onto which the pump and probe beams are focused with 300 mm and 150 mm lenses, respectively.

### Measuring the diffracted light

To measure the angular dependent spectrum of the diffracted probe beam we use a 4f-based detection system (see Supplementary Fig. 1b) to collect the entire angular spectrum before spatially filtering the light at the Fourier plane of the first lens ($f = 100$ mm). The filtered light is then focused by a second lens ($f' = 50$ mm) onto a fibre coupled spectrometer (Princeton Instruments NIRvana infrared spectrometer). The hyperspectral data is acquired by taking a pump-probe delay scan for every position of the spatial filter. The presented data is taken for the delay with largest diffraction efficiency.

### Increasing the duration of the probe pulse

Additionally, the probe beam is passed through a grating, 4-f system with a variable width spatial filter at the Fourier plane. By changing the width of this spatial filter, we can apply a top-hat, spectral filter of variable width to our probe pulse. With this system we stretch our probe pulse to a duration of 690 fs and 480 fs for the experiments undertaken as part of Figs. 2, 3, respectively.

### Modelling the response of ITO

To understand the nonlinear optical behaviour of the ITO film under intense femtosecond laser pumping, the study employs a time-dependent Drude model. This model attributes the observed non-linearity to variations in the effective carrier mass[20], leading to a time-varying plasma frequency. The theory assumes Newton's second law for carriers with a time-dependent mass and constant damping rate, resulting in a modified expression for the polarisation and, consequently, a dynamic permittivity ($\epsilon(t, \omega)$),

$$\epsilon(t, \omega) \sim \epsilon_\infty - \frac{\omega_p(t)^2}{\omega(\omega + i\gamma)}.$$

Here, $\omega_p(t)$ is the phenomenologically time-varying plasma frequency, $\epsilon_\infty$ is the background permittivity and $\gamma$ is the damping constant. This formulation captures the material's non-instantaneous response to the electric field and is used as the basis for simulating how space-time modulations in permittivity diffract incident probe light.

### Modelling space-time permittivity changes

The plasma frequency $\omega_p(x, t)$ is modelled as a convolution of the pump intensity $I(x, t')$ with a temporal response kernel with the form,

$$\tau(t - t') = \frac{1}{2N}\left[1 + \tanh\left(\frac{t - t'}{t_{rise}} - 1\right)\right] e^{-\left(\frac{t - t'}{t_{decay}} - 1\right)}.$$

To model our sample, we use a rise time, $t_{rise}$ of 10 fs and decay time, $t_{rise}$ of 210 fs. Throughout the numerical simulations we use $\epsilon_\infty = 3.9$ and $\gamma = 130$ THz, as well as the unmodulated plasma frequency, $\omega_{p,i} = 450.5$ THz, with a maximum shift in plasma frequency, $\Delta\omega_p = -0.1\omega_{p,i}$. These material parameters are generally based upon similar studies of ITO Moreover, in Supplementary Fig. 5 we demonstrate that these choices of values appropriately model our experiment

and that moderate changes produce a minor impact on the numerically simulated diffraction patterns.

### Numerical simulation methodology

An operator-based approach is adopted to simulate light propagation through the multilayer system composed of Au, ITO, and SiO$_2$. This involves first constructing an operator representation of the space-time dispersive permittivity that acts upon the frequency and wavevector of the electric field distribution[27]. Using this formulation, modified Maxwell's equations are solved to obtain the electric and magnetic field components within the structure. A generalised transfer matrix method is then applied, extended to incorporate operator-valued permittivity, enabling the calculation of reflection and transmission operators for each individual layer and for the full multilayer stack. This framework allows for the computation of the diffracted light spectrum as a function of angle and frequency, which shows strong agreement with the experimental hyperspectral measurements.

### Approximate Theory of Space-Time Diffraction

To support interpretation of the diffraction patterns, we developed an approximate analytical model based on space-time inclined reflectivity profiles. In the single-pump case, the modulation is modelled as a moving reflective patch with velocity $v = c/\sin(\theta_p)$, leading to a Doppler-like relationship between frequency shift and diffraction angle:

$$\frac{\Delta\omega}{\omega} = \frac{v\cos(\theta_p)}{c - v\sin(\theta_p)}\delta\theta.$$

In the configuration with two pump pulses, the modulation consists of two spatially and temporally separated reflective patches. Their interference generates fringes in frequency-momentum space, with the fringe maxima occurring where the phase difference between the scattered waves from each patch is stationary. This condition yields:

$$\Delta k \Delta x - \Delta\omega\Delta t = 0.$$

This equation describes a linear relationship between momentum and frequency shifts across the diffraction pattern. Rearranging this expression and expanding the momentum shift in terms of angle gives the same expression as for the case with a single-pump pulse, however $v = \Delta x/\Delta t$.

The frequency shift can be understood as a Doppler shift. In its rest frame, where the modulation appears to be purely spatial, the probe reflects at a constant frequency. Transforming the reflected wave back to the lab frame, the expression for the scattered frequency ($\omega sc$) yields the same linear relation between frequency shift and scattering angle, given in Eq. (1). Interestingly this Doppler shift theory can be extended to the regime of superluminal modulations (v > c), by performing a complex Lorentz transformation into a frame where the modulation appears to be purely temporal, recovering the same expression for frequency shifts, again agreeing with the analytical Fourier model.

## Data availability

Source data are available for this paper and are available via a public repository at Harwood, Anthony (2024). Space-Time Optical Diffraction from Synthetic Motion. figshare. Dataset. https://doi.org/10.6084/m9.figshare.27925419.v3.

## Code availability

The code used in this study is available at https://doi.org/10.24433/CO.4907533.v2.

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

## Acknowledgements

We would like to thank Euan Hendry for useful discussions and Peter Davies for his help with the data acquisition software. This work was supported by The Engineering and Physical Sciences Research Council (EPSRC), grant number EP/Y015673/1 (A.C.H., S.V., T.V.R., C.H., J.B.P., S.A.H., R.S.), The Val O'Donoghue Scholarship in Natural Sciences (A.C.H.), The Royal Society and TATA for financial support through Grant No. URF\R\211033. (S.A.R.H., C.H.), and The Gordon and Betty Moore Foundation (J.B.P.).

## Author contributions

Conceptualisation: R.S., S.V., S.A.R.H., A.C.H. Methodology: R.S., S.V., A.C.H., S.A.R.H., J.P.B. Investigation: A.C.H., S.V., T.V.R., C.H., R.T., F.W. Visualisation: R.S., S.V., A.C.H., S.A.R.H. Funding acquisition: R.S., S.A.R.H., J.B.P., S.M. Supervision: R.S., S.A.R.H., S.V. Writing—original

draft: R.S., S.A.R.H., S.V., A.C.H. Writing—review & editing: A.C.H., S.V., T.V.R., C.H., R.T., S.M., J.B.P., S.A.R.H., R.S.

## Competing interests

The authors declare no competing interests.
