## [Transparent Peer Review file · Nature Communications]

Space-Time Optical Diffraction from Synthetic Motion

Corresponding Author: Mr Anthony Harwood

This manuscript has been previously reviewed at another journal. This document only contains information relating to versions considered at Nature Communications. Mentions of the other journal have been redacted. Mentions of the other journal have been redacted.

Version 0:

Reviewer comments:

Reviewer #1

(Remarks to the Author)

The current manuscript was previously reviewed by **[Redacted]**, and transferred to Nature Comm. Although the authors have addressed my previous comments at some levels, I believe there are still some ambiguous points that need to be addressed :

1. The authors may consider creating better schematics to show your idea of synthetic motion in the space-time domain in Fig. 1, there's not much difference in Fig. 1a&b, and I can't get the point of such synthetic motion through modulation out of the current illustrations. Maybe move some of the figures in Fig. 2 to Fig. 1?
2. Why ITO? besides its ENZ properties, can this temporal modulation scheme generally apply to other material platforms, even without ENZ ?
3. For Young's double slits experiments, the interference patterns should be scaled with the gap/spacing between the two slits, to make such a case more convincing, is it possible to demonstrate such an effect? at least, numerically?
4. Lastly, for those cases faster than V_p , is there any signature of shock waves or Cherenkov radiation?

(Remarks on code availability)

Reviewer #2

(Remarks to the Author)

The authors have addressed my comments and I am happy to recommend it for publication.

(Remarks on code availability)

Reviewer #3

(Remarks to the Author)

The authors have provided a detailed response to all the points raised in my previous review and clarified those points in the revised manuscript. With these clarifications, I believe the results and analysis are clear, rigorous, and thorough. The experimental results in this work demonstrate an approach to generate diffraction patterns that are tunable in frequency and momentum, which has the potential to controllably drive complex spatio-temporal transformations of light. I believe this work will be of significant interest and warrant publication in Nature Communications.

(Remarks on code availability)

Reply to reviewer feedback for Nature Communications manuscript – “Space-Time Optical Diffraction from Synthetic Motion”

We thank the reviewers and editor for their thoughtful feedback, which helped us clarify and improve the manuscript. All comments have been carefully addressed in this revision.

Below, we provide the reviewers' comments in full, followed by our detailed responses and the corresponding corrections implemented in the manuscript.

REVIEWER COMMENTS

Reviewer #1 (Remarks to the Author):

The current manuscript was previously reviewed by [Redacted], and transferred to Nature Comm. Although the authors have addressed my previous comments at some levels, I believe there are still some ambiguous points that need to be addressed :

1. The authors may consider creating better schematics to show your idea of synthetic motion in the space-time domain in Fig. 1, there's not much difference in Fig. 1a&b, and I can't get the point of such synthetic motion through modulation out of the current illustrations. Maybe move some of the figures in Fig. 2 to Fig. 1?

We thank the reviewer for highlighting the room to improve the clarity of Fig. 1. We have kept the content of Fig. 1a and b the same to illustrate the difference between separable transformations, as well as their linear combinations, and non-separable transformations, which can only be investigated by generating moving permittivity structures. In order to reinforce this, we have added subtitles for panels a and b. Fig. 1c has been altered to match the style of Fig. 2b and illustrates that a permittivity modulation ($\Delta\varepsilon$) structured non-separably in space-time, can lead to synthetic motion with an apparent velocity larger than the speed of light.

The figure caption has also been amended to reflect these changes.

“Fig. 1: Space-time diffraction from a synthetically moving modulation. (a) Separable modulations of the dielectric permittivity ϵ can only induce transformations of the optical spectrum parallel to either the frequency or momentum axes. (b) Meanwhile, the general non-separable modulations of $\epsilon(r, t)$, demonstrated here, can perform arbitrary momentum-frequency transformations that are unobtainable with static metasurfaces. (c) The synthetic motion of permittivity modulations, $\Delta\epsilon(r, t)$, can describe arbitrary trajectories in space-time (solid blue line), even moving at superluminal velocities outside the light cone.”

2. Why ITO? besides its ENZ properties, can this temporal modulation scheme generally apply to other material platforms, even without ENZ ?

ITO enables sub-picosecond, near-unity modulations in its reflectivity, [*Nat. Phys.* 1–4 (2023)]. In principle, the temporal modulation scheme that we demonstrate in this paper is a general one and will work for any medium with the ability to be modulated in ultrafast timescales, for example other transparent conductive oxides as AZO [Kinsey, N. *et al. Optica* 2, 616 (2015).]. Other materials, e.g: semiconductors far from ENZ, could also work, but with low modulation efficiency, $\sim 10^{-6}$ - 10^{-3} , and therefore with much lower space-time diffraction efficiency.

3. For Young’s double slits experiments, the interference patterns should be scaled with the gap/spacing between the two slits, to make such a case more convincing, is it possible to demonstrate such an effect? at least, numerically?

We appreciate the reviewer’s suggestion regarding an experimental demonstration of the inverse relationship between fringe separation and slit spacing. That is beyond the scope of this work and will be the subject of a future work.

A comprehensive study of how fringe separation depends on the space-time separation of the slits, supported by detailed numerical simulations, is a substantial undertaking that we believe warrants a dedicated investigation.

We have completed the following analytical simulation of the fringe separation with space/time period of the two discretely modulated slits in our experiment. The conclusion of these simulations is that the spacing along k and ω depends on the separation of slits in space and time, respectively. This can also be seen from Eq. (46) in the SI. Since the numerical simulations are in turn derived from this analytical theory, we believe this demonstration should address the reviewer’s concern.

4. Lastly, for those cases faster than V_p , is there any signature of shock waves or Cherenkov radiation?

In our paper we do not measure Čerenkov radiation or signatures of shock waves as predicted for example by [Sloan, J. *et al. Nat. Phys.* **18**, 67–74 (2022)]. We thank the reviewer for recognising that our experimental system represents a promising route for detecting Čerenkov radiation generated in vacuum. Čerenkov radiation is very weak, which makes it challenging to detect, and comes at specific emission angles. This is the focus on future experiments that we are planning.

Reviewer #2 (Remarks to the Author):

The authors have addressed my comments and I am happy to recommend it for publication.

We would like to thank Reviewer #2 for their thoughtful feedback and appreciate their recommendation for publication.

Reviewer #3 (Remarks to the Author):

The authors have provided a detailed response to all the points raised in my previous review and clarified those points in the revised manuscript. With these clarifications, I believe the results and analysis are clear, rigorous, and thorough.

The experimental results in this work demonstrate an approach to generate diffraction patterns that are tunable in frequency and momentum, which has the potential to controllably drive complex spatio-temporal transformations of light. I believe this work will be of significant interest and warrant publication in Nature Communications.

We would like to thank Reviewer #3 for their thoughtful feedback and appreciate their recommendation for publication.